# Dissipation of Eutrophic Substances in Grass Carp Aquaculture Pond Water by Ozone

**Zhe Chen** [1,2], **Xingguo Liu** [2,*], **Xiangyu Cheng** [1,2] **and Zeyu Guo** [1,2]

[1] College of Fisheries and Life Science, Shanghai Ocean University, Shanghai 201306, China; chenzhe19991223@163.com (Z.C.); cxy991224@163.com (X.C.); 17861510083@163.com (Z.G.)

[2] Fishery Machinery and Instrument Research Institute, Chinese Academy of Fishery Sciences, Shanghai 200092, China

[*] Correspondence: liuxingguo@fmiri.ac.cn; Tel.: +86-13301856629

**Abstract:** Suspended solids (SS) have become the main cause of water quality deterioration in aquaculture ponds. The application of ozone to aquaculture water bodies can improve the water quality and facilitate the removal of suspended solids. We used different concentrations of ozone to oxidize aquaculture water and then determined the resulting concentration of suspended solids and their particle size distribution, as well as CODMn, UV254, algal density, and nitrogen content. The results showed the following: (i) The decrease in SS was the highest, amounting to 10.47%, after the addition of 2 mg/L of ozone and the completion of the oxidation reaction. Ozone caused the fragmentation of large SS particles and the flocculation of small particles, and these effects became more pronounced as the ozone concentration increased; (ii) After the introduction of ozone, the humus macromolecules naturally present in the water were oxidized, which improved the biochemical degradation of the water pollutants; (iii) Ozone oxidation caused the degradation of algae, resulting in a decrease in phytoplankton biomass and in the eutrophication of the water body; (iv) As the ozone concentration increased, the level of nitrous nitrogen decreased, while the concentrations of nitrate nitrogen and ammonia nitrogen were unchanged. The highest increase in total nitrogen was measured when 0.5 mg/L of ozone was applied. This study provides a theoretical basis for the application of ozone to eliminate eutrophic substances in freshwater ponds.

**Keywords:** ozone oxidation; suspended solids; eutrophication; grass carp pond

## 1. Introduction

In recent years, aquaculture has been developing rapidly, accompanied by environmental problems such as water pollution [1,2] caused by the casual discharge of aquaculture tail water. Pond culture is one of the most important methods of freshwater aquaculture. However, the blind pursuit of increasingly higher yields and the lack of supporting technology for the management of intensive aquaculture may cause the rapid deterioration of pond aquaculture water quality, the frequent occurrence of diseases, the increase of pollution discharge [3], and the decline in the quality of production, posing serious challenges to the development of this practice [4]. Studies have shown that in high-density intensive aquaculture systems, about 25–30% of the feed in the form of suspended solids remains in the water system. These suspended solids carry various nitrogenous and phosphorous substances, whose prolonged persistence in the water [5] will lead to an increase in ammonium nitrogen ($NH_4^+$-N), nitrate nitrogen ($NO_3^-$-N), nitrite nitrogen ($NO_2^-$-N), organic matter, and phosphorus (P) in the aquaculture system. The enrichment of nitrogen and phosphorus in bodies of water can lead to eutrophication and algal blooms that seriously affect aquatic ecosystems [6]. $NO_2^-$-N causes a decrease in the blood's ability to transport oxygen in cultured fish, leading to respiratory distress and, in severe cases, physiological hypoxia and death. SS also cause hazards such as gill damage [7], accumulation of harmful substances, and toxic effects in cultured fish, and promote pollution in the surrounding

water environment [8]. Therefore, effective solid particulate management tools are needed for the timely removal of suspended matter from the aquaculture systems [9,10].

Currently, the removal of suspended solids (SS) is effective in factory-based recirculating aquaculture systems (RAS). The most commonly used methods to remove suspended solids include mechanical filtration, gravitational sedimentation, and air flotation, which are utilized depending on the characteristics of the suspended matter in the body of water, such as density, particle size, etc. [11,12]. A generalized classification method for aquaculture solids, i.e., the determination of their particle size distribution (PSD), was introduced for the first time by Rueter et al. [13]. The PSD has been classically used as a characterization parameter for SS, alongside other indicators such as total suspended solids (TSS). The overall removal efficiency of the above-mentioned techniques used to remove SS depends on the PSD of suspended solids [14]. In fact, solids larger than 100 μm can be effectively removed by gravity or mechanical filters, while suspended solids smaller than 50 μm are difficult to remove economically. Unfortunately, more than 95% of the suspended solids in aquaculture waters are less than 20 μm in diameter.

Additional methods are used in aquaculture water treatment. These include biological treatments such as the use of an activated sludge and chemical oxidation. The latter has advantages such as rapidity and limited cost [15]. Studies have shown that the application of ozone to aquaculture water can effectively inhibit pathogenic microorganisms in aquaculture systems of fish, shrimps, crabs, shellfish and other aquatic animals and promotes the removal of suspended solids, organic wastes, and the oxidation of $NO_2^-$-N and $NH_4^+$-N. Ozone reaction in aquaculture water is rapid and results in a very small production of harmful by-products. In addition, as oxygen is the end product of the reaction, this method will increase the amount of dissolved oxygen in the body of water [16,17]. In wastewater treatment, ozone oxidizes pollutants by direct oxidation or indirect oxidation. Direct oxidation is a direct reaction between ozone and pollutants; indirect oxidation requires ozone-induced decomposition and the generation of hydroxyl radicals, followed by the oxidation of organic matter. In direct oxidation, ozone molecules selectively react with pollutants, converting macromolecules into small molecules of organic matter. The overall degree of oxidation is not high, but macromolecules are broken into small molecules of organic matter which usually have various biochemical properties. In indirect oxidation, the produced ·OH radicals are very effective oxidants in advanced oxidation that can quickly oxidize and even mineralize the organic matter present in the water. This oxidation process is not selective but is effective on a wide range of difficult-to-degrade organic compounds. Therefore, ozone can decompose relatively non-biodegradable organic compounds into smaller and biodegradable compounds in aquaculture water [18,19]. Because of its strong oxidizing properties, ozone can quickly degrade algal cells, inhibiting algal bloom [20], but the removal efficacy of different algae varies greatly, mainly because the larger the algal cell volume, the thicker the cell membrane, and the more abundant the extracellular material, the higher the algae tolerance to ozone. Theoretically, the application of ozone to freshwater aquaculture ponds could successfully remove suspended particulate matter, algae, organic compounds, $NO_2^-$-N, and $^+$-N.

In fact, ozone has been widely used in aquaculture to achieve disinfection and improve the water quality, generally in combination with other treatments, such as bubble bath separation to remove fine suspended particles [21], UV irradiation to achieve the full-flow disinfection of recirculating water [19], and UV irradiation together with chlorine application to remove resistant organisms in the body of water [22,23]. A large number of studies have shown that ozone can effectively remove major pollutants from aquaculture water such as SS, $NO_2^-$-N, and organic matter. However, most of them were carried out in seawater RAS and freshwater RAS. There is a relative lack of research on the use of ozone to treat freshwater aquaculture ponds, which are different from the RAS, as they contain abundant zooplankton and algae. Therefore, the effect of ozone treatment in freshwater aquaculture ponds is not yet clear. This study was carried out to evaluate the efficacy of ozone in the removal of suspended particulate matter from aquaculture ponds and

identify its shortcomings. The goal of this study was to provide a theoretical basis for the application of ozone to pond aquaculture water to remove suspended particles and dissolve the eutrophic substances present in the body of water.

## 2. Materials and Methods

### 2.1. Materials

Typical grass carp aquaculture ponds in the San Mao Aquaculture Farm ($30°57'1.89''$ N, $121°08'52.21''$ E) in Songjiang District, Shanghai were selected. The ponds were 100 m × 50 m in size, with an average water depth of 1.6 m and a water area of 0.47 $hm^2$; 4000 grass carp were cultured in the ponds.

Samples were collected according to the five-point sampling method, using a 5 L water sampler to sample the water body of the grass carp breeding ponds, mixing the water samples at each sampling point, and then placing the mixed water samples in a 2 L ozone reaction container.

In this experiment, a high concentration of pure oxygen(Shanghai Wetry Standard Gas Analysis Technology Co., Ltd., Shanghai, China) was used as the gas source to produce a high concentration of ozone (in a 2 L ozone) reaction vessel (height of 195 mm, diameter of 138 mm), and the concentration of ozone passed into the reaction vessel was set to be 0.0 mg/L, 0.5 mg/L, 1.0 mg/L, 1.5 mg/L, 2.0 mg/L, 2.5 mg/L (considering that ozone and its residues can be toxic to farmed fish, ozone concentrations within a safe and controllable range were chosen for the experiment). Ozone was passed through the aeration stone into the reaction vessel. To characterize the amount of ozone dissolved in the water column, we determined the total residual oxidant (TRO) concentration using the DPD (N,N-diethyl-p-phenylenediamine) colorimetric method and a portable detector LH-CE10F (Zhejiang Lohand Environment Technology Co., Ltd., Shanghai, China). The value of TRO concentration indicated the amount of ozone remaining in the water column. The TRO concentration in the water column after 30 min of the reaction at each concentration was 0; therefore, the reaction was completed in 30 min. Then, the water samples were analyzed to determine the particle size distribution (PSD) and various water quality indicators.

### 2.2. Methods

For PSD determination, we used a BT-9300SE laser particle sizer, with a measurement range of 0.1~1000 μm, which allowed for the calculation of the volume distribution and the quantity of SS.

The SS concentration in the body of water was determined using the national standard method GB17378.4-1998. Specific methods are briefly described below.

A Whatman GF/F glass fiber filter membrane (0.7 μm) at 100~105 °C was dried for over 2 h. The SS amount A1 (mg), obtained after filtration of a water volume, was dried at 100~105 °C for 4 h. After cooling, the amount $A_2$ (mg) was obtained. Then, the membrane was burned at 550 °C for 2 h, and, after cooling, the SS mass $A_3$ (mg) was obtained. The mass concentration of SS(A) was calculated according to the following equation:

$$A = (A_2 - A_1)/V \tag{1}$$

$A_{(OSS)}$ was determined by

$$A_{(OSS)} = (A_2 - A_3)/V \tag{2}$$

The algae density was determined by counting under a Olympus CX31 microscope. We added 15 mL of Luger's reagent to 1 L of a test sample, placed the sample in a dark place for 24 h, then concentrated it to about 40 mL using a siphon. The phytoplankton samples were identified under a 10 × 40 microscope, the volume of phytoplankton was determined using a 0.1 mL counting frame, and the biomass of phytoplankton was calculated by a volumetric method.

The algal density was calculated as

$$N = \frac{C_s}{F_s \times F_n} \times \frac{V}{U} \times P_n \tag{3}$$

where N is the density of phytoplankton cells (or individuals) in 1 L of water, unit: cells/L; $C_s$ is the area of the counting frame, unit: $mm^2$; $F_s$ is the area of the field of view, unit: $mm^2$; $F_n$ is the number of counted fields of view, unit: one; V is the sample volume obtained from 1 L of water after sedimentation and concentration, unit: mL; U is the volume of the counting frame, in mL; and $P_n$ is the number of phytoplankton cells (or individuals) counted per slice.

The biomass was calculated as

$$PB = \sum_{i=1}^{k}(V_i \times N_i) \times 10^{-9} \tag{4}$$

where PB is the biomass of phytoplankton (mg/L), $V_i$ is the average volume of alga i ($\mu m^3$), and $N_i$ is the density of alga i (cells/L).

$UV_{254}$ treatment: The water samples were filtered through a 0.45 μm membrane, and their ultraviolet absorbance ($UV_{254}$) was determined by ultraviolet spectrophotometry using a 1 cm quartz cuvette with a unit cuvette optical range at a wavelength of 254 nm, according to

$$UV_{254} = (A/b) \times D \tag{5}$$

where b is the optical range of the cuvette used in cm; A is the measured absorbance value; and D is the dilution factor, related to the dilution with pure water (=final water sample volume/initial water sample volume).

The total nitrogen was determined by the alkaline potassium persulfate digestion method (HJ636-2012 [24]); ammonia nitrogen was determined by nano reagent photometry (HJ535-2009 [25]); nitrite nitrogen was determined by molecular absorption spectrophotometry (GB7493-87 [26]); nitrate nitrogen was determined by ultraviolet spectrophotometry (HJ/T346-2007 [27]); and the chlorophyll a concentration was determined by the hot ethanol extraction–spectrophotometric method (HJ897-2017 [28]).

## 3. Results

### 3.1. SS

The trend of variation of the SS concentration after the oxidation of aquaculture water using different ozone concentrations is shown in Figure 1. The original concentration of SS was 76.40 mg/L, of which the concentration of organic suspended solids (OSS) was 74.80 mg/L, accounting for 97.91%. It was found that the SS content showed an overall decreasing trend with the increase in the ozone fluence. The rate of SS concentration decrease was 3.14% after the reaction of 0.5 mg/L of ozone ($P < 0.05$), 6.81% when using 1.5 mg/L of ozone, and 10.47% in the presence of 2 mg/L of ozone. The SS concentration increased when the ozone concentration was 2.5 mg/L. This could be due to experimental error, or it could be that the ozone flocculated the small particles, and the small particles that originally passed through the membrane were retained, but it is highly unlikely. The proportion of organic SS in the water body gradually increased with the increase in ozone concentration. The highest percentage of OSS was 99.44%, measured in the presence of an ozone concentration of 2.5 mg/L.

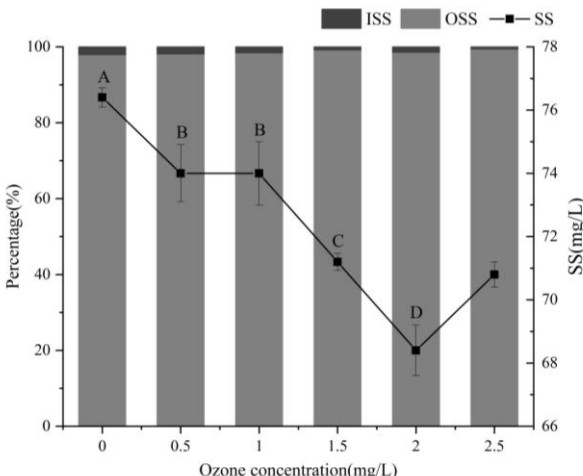

**Figure 1.** Variation of SS concentration in water treated with different ozone concentrations. (Different capital letters indicate significantly different SS concentrations across ozone treatments at different concentrations at the 0.05 probability level, same as below).

The passage of ozone through the aquaculture water influenced the particle size distribution (PSD) of SS (Figure 2). After treatment with 0.5 mg/L of ozone, the volume of SS with diameter > 200 μm was reduced from 0.49% to 0%, and that of SS with diameter of 100~200 μm was reduced from 6.07% to 1.14%; the latter was further reduced to 0.78% after oxidation with 2.5 mg/L of ozone. Ozone oxidation induced the decomposition and fragmentation of the larger SS particles (100~1000 μm). After treatment with 0.5 mg/L of ozone, SS with a diameter of 1~10 μm increased from 20.6% to 23.12% in the original sample; a further increase to 29% was measured after treating with an ozone concentration of 2.5 mg/L. This indicated that ozone fragmented the large SS particles (100~1000 μm) into small particles (1~10 μm). In terms of quantity distribution, the content of SS > 50 μm in the original sample amounted to 0%, and that of SS of 0~1 μm decreased from 91.63% to 90.32% after treatment with 0.5 mg/L of ozone, and further to 87.63% after treatment with 2.5 mg/L of ozone. In addition, the content of SS of 1~10 μm was 8.27% in the original sample and increased to 9.58% after treatment with 0.5 mg/L of ozone and to 12.23% in the presence of 2.5 mg/L of ozone. Therefore, after ozone treatment, the number of small SS particles (0~1 μm) decreased, and the number of large particles (1~10 μm) increased, which indicated a flocculation effect of ozone. Nowadays, there is not a definite explanation about the principle of why ozone has a flocculation effect on SS, and further research is expected.

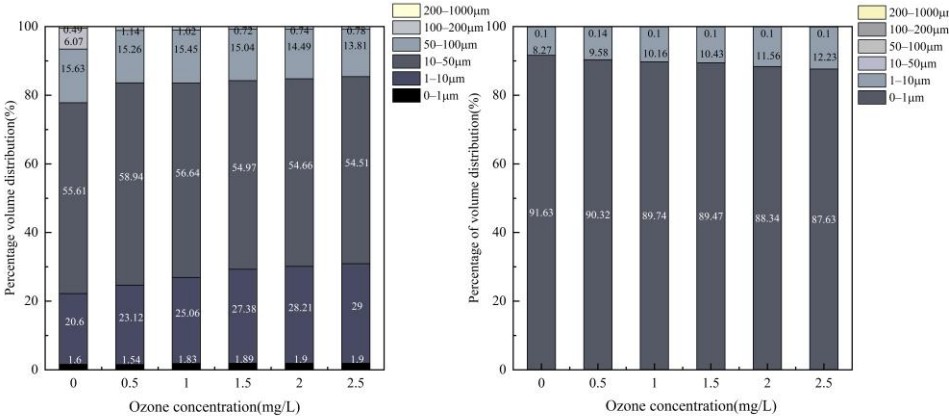

**Figure 2.** Changes in volume distribution and amount distribution of suspended particulate matter according to their particle size after treatment with different ozone concentrations.

### 3.2. Water Quality Indicators

We then investigated the changes of water quality indexes, such as $COD_{Mn}$, $UV_{254}$, $NO_2^-$, $NO_3^-$, $NH_4^+$, and TN, after complete water oxidation by ozone. The changes in $COD_{Mn}$ and $UV_{254}$ are shown in Figure 3. With the gradual increase of the ozone concentration, $COD_{Mn}$ showed irregular changes, while the $UV_{254}$ values constantly decrease. When the ozone concentration was 0.5 mg/L, the $COD_{Mn}$ concentration decreased slightly from 88 mg/L to 86 mg/L; in contrast, with the increase in ozone concentration, the $COD_{Mn}$ concentration increased, reaching the value of 97 mg/L when the ozone concentration was 1.5 mg/L and that of 95 mg/L when the ozone concentration was 2.5 mg/L. The irregular changes of $COD_{Mn}$ are shown in Figure 3. The irregular changes of $COD_{Mn}$ indicated that the amount of reducing substances to be oxidized in the body of water changed irregularly after ozone was introduced. This could be due to the fact that ozone acted on the difficult-to-degrade organic matter in the water, improving the water biochemistry. As the amount of ozone introduced increased, the biochemically weak organic molecules were captured, leading to an increase in $COD_{Mn}$ values, while ozone oxidized the organic molecules, leading to a decrease in $COD_{Mn}$ values, thus resulting in irregular changes in $COD_{Mn}$ values. The initial value of $UV_{254}$ was 0.223 $cm^{-1}$, which decreased with the increase in ozone concentration. At an ozone concentration of 2.5 mg/L, $UV_{254}$ decreased to 0.144 $cm^{-1}$, a decrease of 35.43% ($P < 0.05$). The decreasing $UV_{254}$ indicated a continuous decrease in the amount of naturally occurring humus-like macromolecular organic matter, as well as of aromatic compounds containing C=C double bonds and C=O double bonds in the ozonated water. These results may be attributed to the oxidizing effect of ozone on the macromolecular organic matter in the water, resulting in the partial removal of the organic matter by decomposition.

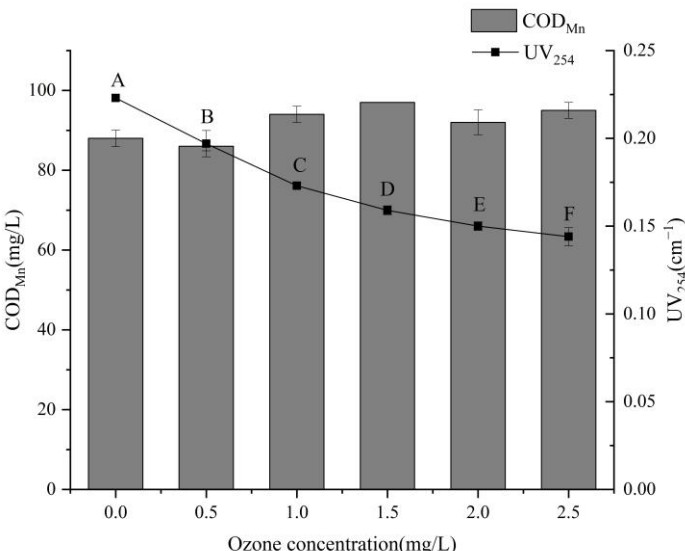

**Figure 3.** Variation of $COD_{Mn}$ and $UV_{254}$ after treatment with different ozone concentrations.

The changes in the concentrations of $NO_2^-$, $NO_3^-$, $NH_4^+$, and TN are shown in Figure 4. $NO_2^-$ concentration was low in the original sample, only 0.006 mg/L, and decreased to 0 mg/L immediately after ozone was introduced. This may be due to the fact that ozone easily oxidized $NO_2^-$ because of its low initial amount; if the concentration of $NO_2^-$ in the water body was higher, it might decrease linearly with the increase in ozone concentration. $NO_3^-$ concentration in the original sample was 0.28 mg/L and increased after the complete reaction with ozone, reaching a value of 0.57 mg/L, which corresponded to an increase of 103.75% in the presence of the ozone concentration of 2.5 mg/L. The concentration of $NH_4^+$ in the original sample was 0.41 mg/L and increased after the ozone treatment, reaching a value of 0.66 mg/L in the presence of 0.5 mg/L of ozone and further increasing to 1.26 mg/L in the presence of 2.5 mg/L of ozone, with

a rate of increase of 203.75% ($P < 0.05$). The concentration of TN was 1.78 mg/L in the original sample and increased after ozone treatment. The change trend was the same as that observed for $NH_4^+$ and $NO_3^-$. When the ozone concentration was 0.5 mg/L, the TN concentration was 3.19 mg/L ($P < 0.05$), while the ozone concentration was 2.5 mg/L, the TN concentration was 3.44 mg/L, with an increase of 93.49% ($P < 0.05$). The increase in the concentrations of $NO_3^-$, $NH_4^+$, and TN indicated that ozone oxidized the reducing N-containing macromolecules in the water, decomposing them and then releasing N. In summary, the increase in the $NO_3^-$, $NH_4^+$, and TN concentrations was caused by the release of N from the decomposition of macromolecular organic matter.

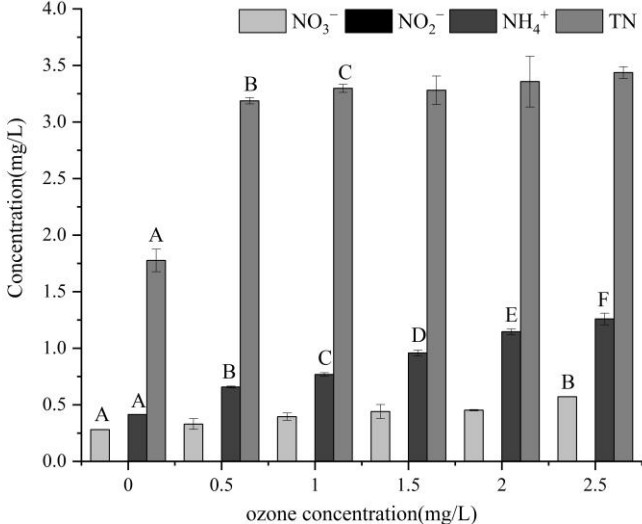

**Figure 4.** Changes in nitrogen concentration after treatment with different ozone concentrations.

### 3.3. Algal Indicators

We then explored the changes in algal density, chlorophyll a (Chla) concentration, and biomass in the water column after treatment with different ozone concentrations. The changes in algal density are shown in Table 1. We found that the algal density decreased continuously after ozone treatment, and the decrease was greater as the ozone concentration increased. The decrease in algal density indicated that ozone could rapidly and efficiently inactivate the algae. In particular, the inactivation of *Microcystis aeruginosa* was the most obvious.

**Table 1.** Algae density after treatment with different ozone concentrations.

| Ozone Concentration (mg/L) | *Microcystis* ($10^7$ cells/L) | *Melosira* ($10^6$ cells/L) | *Scenedesmus* ($10^6$ cells/L) | *Anabaena* ($10^6$ cells/L) | *Euglena* ($10^5$ cells/L) |
|---|---|---|---|---|---|
| 0 | 3.52 | 6.04 | 6.90 | 5.44 | 4.41 |
| 0.5 | 2.11 | 3.69 | 4.73 | 2.53 | 2.21 |
| 1.0 | 2.04 | 3.43 | 4.15 | 2.09 | 1.60 |
| 1.5 | 1.93 | 3.39 | 3.89 | 1.81 | 1.79 |
| 2.0 | 1.86 | 3.37 | 3.75 | 1.79 | 1.81 |
| 2.5 | 1.30 | 2.91 | 3.69 | 1.82 | 1.00 |

The congeneric algae that were present in low amounts in the samples are grouped together in Table 1. *Microcystis* is dominated by *Microcystis aeruginosa*; *Melosira* is dominated by *Melosira granulate* and *Melosira varians*; *Scenedesmus* includes *Scenedesmus abundans*, *Scenedesmus carinatus*, *Scenedesmus dimorphus*, and *Scenedesmus quadricauda*; *Anabaena* includes *Anabaena circinalis* and *Anabaena variabilis*; *Euglena* includes *Euglena caudata*, *Euglena polymorpha*, *Euglena sangguinea*, *Euglena wangi*, and *Euglena spirogyra*.

The variation of phytoplankton biomass and Chla concentration with the ozone concentration is shown in Figure 5. It was found that the phytoplankton biomass decreased most drastically after the 0.5 mg/L ozone treatment, from 55.96 mg/L to 32.91 mg/L, with a decrease of 41.19%. The biomass decrease slowed down with the increase in ozone flux,

reaching the value of 23.28 mg/L, which corresponded to a decrease of 58.40% with respect to the original amount, after treatment with 2.5 mg/L of ozone. This could be due to the different removal rates of different algal species, as ozone easily inactivated and lysed algae with low ozone tolerance and thin cell membranes, whereas it was less efficient on algae with a thicker cell membrane and more extracellular material, which were more difficult to remove.

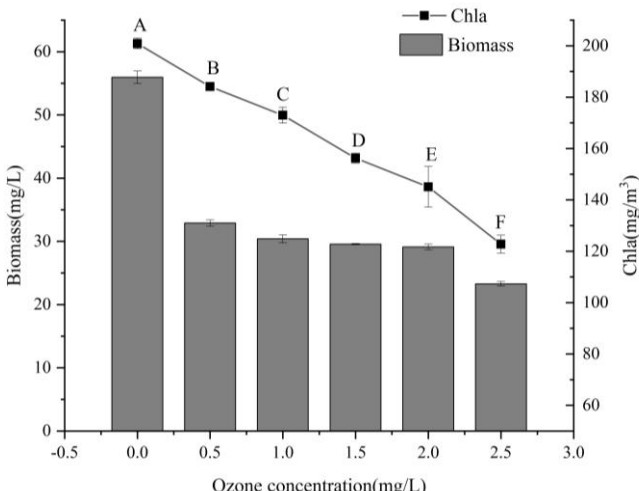

**Figure 5.** Changes in phytoplankton biomass and Chla concentration after treatment with different ozone concentrations.

As the concentration of ozone increased, the concentration of Chla decreased, and the color of the ozonated water changed from yellow to green during the test. In the original sample, Chla concentration was 200.88 mg/m$^3$ and decreased to 184.14 mg/m$^3$, i.e., by 8.33% after the complete reaction of 0.5 mg/L of ozone. When the ozone concentration was 2.5 mg/L, Chla concentration was 122.76 mg/m$^3$, corresponding to a decrease of 38.89% compared with the concentration in the original water sample ($P < 0.05$). The decrease in the chlorophyll level indicated a decrease in the amount of planktonic algae, which, when eutrophication occurs in the water body, multiply and generate harmful blooms. Ozone could inactivate the algae, effectively alleviating eutrophication and improving the water quality.

## 4. Discussion

### 4.1. Removal of Suspended Particulate Matter

SS in freshwater aquaculture ponds derives mainly from exogenous feed; therefore, the main SS component is organic matter. SS are water pollutants of concern in aquaculture ponds. A number of studies investigated the application of ozone for the removal of SS from aquaculture systems in order to improve the water quality. For example, Manuel et al. [10] and Kim et al. [29] combined ozone with foam fractionation to treat RAS, and Sanni L et al. [21] investigated the application of ozonation in RAS to maintain a high level of water quality in the system. The present study found that ozone oxidation had various effects on SS. We showed that ozone input could reduce the SS concentration in aquaculture water and induced the fragmentation of large SS particles and the flocculation of small particulate matter. Ji Mingdong et al. [30] showed that ozone promoted the flocculation and fragmentation of SS in seawater recirculating aquaculture systems, in agreement with our results. It was found that most of the SS was composed of OSS. Ozone oxidation should also occur in OSS, but the actual test found that ozone did not have a significant effect on the reduction of SS concentration. However, Summerfelt et al. [16] showed that the SS removal rate when using filters was increased by 33% on average after ozone was introduced; in addition, ozone reduced the clogging of the panel of micro-screen filters. This could be due to the fact that ozone can only change SS characteristics, such as particle

size and organic content, without removing them from the water. However, the use of ozone in conjunction with other water treatment methods can enhance SS removal by the chosen treatment equipment, indicating that this combined strategy is the optimal solution for the removal of SS.

*4.2. Impact on Water Quality Indicators of Water Bodies*

The degree of pollution present in aquaculture ponds is shown not only by the SS concentration, but also by the content of organic matter in the water body. The COD index, i.e., chemical oxygen demand (COD), reflects the degree to which water is contaminated by organic substances, and $UV_{254}$ represents the amount of naturally occurring humus-like macromolecules in the water, as well as the amount of aromatic compounds containing the C=C double bond and the C=O double bond. The value of COD was 31.3% after treatment of culture water with ozone in a study by Feng Yang et al. [31]. Summerfelt et al. [16] treated culture water with ozone and reported that the COD was reduced by 40.1%, from the initial 43.6 mg/L to 26.1 mg/L. The changes in $COD_{Mn}$ we found in this study are different from those reported previously, probably because the pond aquaculture water body and the RAS water body have a different composition, with that of the pond water body being more complex, which would lead to different variations in $COD_{Mn}$. Ozone treatment is generally recognized as a potential method for the oxidative elimination of all types of organic impurities in water. This is not consistent with the changes in the $COD_{Mn}$ values reported in this study. Guo Enyan et al. [32] applied ozone to circulating aquaculture wastewater, and the results showed that the final removal rate of $UV_{254}$ reached 34.4%, indicating that ozone could oxidize difficult-to-degrade organic matter into soluble biochemical substances. The present study obtained similar results. Considering the results for $COD_{Mn}$ and $UV_{254}$, it is speculated that ozone treatment oxidizes and decomposes those substances present in the water that are difficult to detect, thus leading to an increase in the $COD_{Mn}$ value. In fact, ozone can oxidize the macromolecular organic matter in the water into small-molecule organic matter, improve the biochemical degradation of the water, and facilitate the purification of aquaculture wastewater.

We found a linear increase in the $NH_4^+$ and $NO_3^-$ concentrations in this study. Glen et al. [33] found that nitrate nitrogen increased with increasing ozone doses and concluded that the linear increase in ammonia nitrogen was caused by the oxidative decomposition of organic matter (amines, humus). This result is consistent with our results, which, combined with the decrease in the $UV_{254}$ values, again suggest that the increase in $NH_4^+$ and $NO_3^-$ could be due to the release of nitrogen from the oxidative decomposition of macromolecules in water by ozone. Song Benben et al. [34] showed that in seawater RAS, ozone was effective in removing $NH_4^+$ and $NO_2^-$ from the system. This is different from the results of this study. It is possible that the increase in $NH_4^+$ and $NO_3^-$ could be due to the different composition of freshwater ponds and RAS water, affecting the changes in the nitrogen fraction. The TN concentration increased the most when the ozone dose was 0.5 mg/L; its subsequent increase was slow, probably due to the fact, already discussed above, that the ozone first acted on difficult-to-decompose compounds, causing their fragmentation and the of release nitrogen substances, which led to a dramatic increase in TN. Guan Chongwu et al. [35] showed that ozone reacts preferentially with organic pollutants in RAS water, followed by $NO_2^-$ and $NH_4^+$. Our results similarly indicate that ozone reacts preferentially with macromolecular organic matter in freshwater pond aquaculture water bodies.

*4.3. Effects on Algae*

Algal bloom in pond aquaculture water bodies is different from that in freshwater RAS water bodies. Algal bloom affects the normal production of aquaculture and should thus be prevented. At present, ozone is widely used for intensive algal removal due to its efficient oxidizing properties. The results of this experimental study showed that ozone could rapidly inactivate algae and reduce the occurrence of algal bloom when introduced

into aquaculture water. We found that ozone tended to kill large algal species first and achieved their removal using low ozone concentrations. Currently, there is controversy concerning the use of ozone alone for algae removal. Zeng Ting et al. [36] reported that, although ozone has a strong oxidizing ability, it cannot completely and thoroughly degrade algae; therefore, it causes the release of organic matter of algal origin and the increase in the concentration of dissolved organic matter, which will affect the quality of water. Sun Wentao [37] suggested that the combination of ozone with the air flotation method can results in the oxidation and degradation of organic pollutants while removing algal cells, thus improving the shortcomings of the traditional process of dissolved air flotation, such as the low the adhesion of the air bubbles to algal cells and the low efficiency in the removal of organic pollutants. Therefore, the combination of ozone with other methods for the removal of pollutants from water bodies will improve the treatment of eutrophic water.

## 5. Conclusions

We showed that ozone treatment of freshwater pond culture water led to: (i) a decrease in SS content; (ii) the fragmentation of large SS particles and the flocculation of small particles; (iii) the oxidation of naturally occurring humus-like macromolecular organic matter in the water, improving the biochemical degradability of the water body; (iv) a decrease in algae density, phytoplankton biomass, and the eutrophication of the water body; and (v) an increase in $NO_2$, $NO_3^-$, $NH_4^+$, and TN concentrations.

This study showed that ozone was effective in removing SS, algae, and $NO_2^-$, decomposing macromolecular organic matter, and improving the biochemistry of the water body. Therefore, it has good prospects for application in the purification of freshwater pond water. However, the use of ozone alone in freshwater ponds may have certain disadvantages, and it is recommended to develop its application in combination with other tailwater treatment methods to improve its shortcomings.

**Author Contributions:** Conceptualization, Z.C. and X.L.; methodology, Z.C.; software, Z.C.; validation, Z.C., X.C. and Z.G.; formal analysis, Z.C.; investigation, Z.C., X.C. and Z.G.; resources, X.L.; data curation, Z.C.; writing—original draft preparation, Z.C.; writing—review and editing, X.L.; visualization, X.L.; supervision, X.L.; project administration, X.L.; funding acquisition, X.L. All authors have read and agreed to the published version of the manuscript.

**Funding:** This research was funded by [National Freshwater Genetic Resource Center] grant number [18537].

**Data Availability Statement:** The data that support the findings of the study are available from the corresponding author upon request.

**Conflicts of Interest:** The authors declare no conflict of interest.

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
