# Peer review of "Dissipation of Eutrophic Substances in Grass Carp Aquaculture Pond Water by Ozone"

_water, doi:10.3390/w15183167_

Round 1

Reviewer 1 Report

1. In the section 2.1, what is the author's basis for collection of samples and setting of ozone concentrations?

2. In the section 3.1, as the further increase of ozone concentrations, the authors should explain why the concentration of SS increased 3.0% after the reaction of 2.5 mg/L ozone.

3. In the section 3.2, the author should discuss in depth why CODMn showed irregular changes with the gradual increase of ozone concentrations.

4. In the section 3.2 (lines 227-232), the author should explain why concentrations of NO2- decreased to 0 mg/L immediately when ozone was introduced

5. In the section 3.3, the author should explain why the ozone is effective in alleviating the deepening of eutrophication in the water body under the condition that concentrations of NO3-, NH4+ and TN were increased after ozone oxidation.

6. In the section 3.1 (lines 197-199), after ozone treatment, the author should explain why the number of small particle size SS (0~1 μm) decreased and the number of large particle size SS (1~10 μm) increased, showing flocculation effect.

7. In the Introduction section (lines 34-42), the authors should discuss deeply about the hazards of pollutants such as nitrogen and phosphorus, so as to emphasize the necessity and significance of removing pollutants. This can help the paper become more complete. The reference (Bioresource Technol. 362 (2022) 127819) maybe helpful for this section and was suggested to be cited.

8. In the Introduction section (lines 60-66), the authors should compare deeply chemical oxidation and other techniques to highlight chemical oxidation with advantages. The reference (J. Clean. Prod. 399 (2023) 136546) maybe helpful for this section and was suggested to be cited.

Author Response

  1. In the section 2.1, what is the author's basis for collection of samples and setting of ozone concentrations?

Sample collection: The water bodies targeted in this paper are freshwater aquaculture ponds, so the samples were collected from farmed grass carp ponds.

Setting the ozone concentration: Some papers were referred to, and since the method used was direct injection, the ozone concentration within a safe and controllable range was chosen for the experiment in consideration of the fact that ozone and its residuals would be toxic to farmed fish.

  1. In the section 3.1, as the further increase of ozone concentrations, the authors should explain why the concentration of SS increased 3.0% after the reaction of 2.5 mg/L ozone.

I am also puzzled by this place, it could be an experimental error, or it could be due to ozone flocculation of small particles, the original small particles that passed through the membrane were trapped, but it is very unlikely, so the text does not give an explanation.

  1. In the section 3.2, the author should discuss in depth why CODMn showed irregular changes with the gradual increase of ozone concentrations.

It has been added to the paper.

  1. In the section 3.2 (lines 227-232), the author should explain why concentrations of NO2- decreased to 0 mg/L immediately when ozone was introduced

It has been added to the paper.

  1. In the section 3.3, the author should explain why the ozone is effective in alleviating the deepening of eutrophication in the water body under the condition that concentrations of NO3-, NH4+ and TN were increased after ozone oxidation.

It has been added to the paper.

  1. In the section 3.1 (lines 197-199), after ozone treatment, the author should explain why the number of small particle size SS (0~1 μm) decreased and the number of large particle size SS (1~10 μm) increased, showing flocculation effect.

It was found that ozone can flocculate small particles of SS, but the principle is difficult to explain and a review of the literature did not reveal a specific reason. Therefore, no explanation is given in this paper.

  1. In the Introduction section (lines 34-42), the authors should discuss deeply about the hazards of pollutants such as nitrogen and phosphorus, so as to emphasize the necessity and significance of removing pollutants. This can help the paper become more complete. The reference (Bioresource Technol. 362 (2022) 127819) maybe helpful for this section and was suggested to be cited.

It has been added to the paper.

  1. In the Introduction section (lines 60-66), the authors should compare deeply chemical oxidation and other techniques to highlight chemical oxidation with advantages. The reference (J. Clean. Prod. 399 (2023) 136546) maybe helpful for this section and was suggested to be cited.

It has been added to the paper.

Reviewer 2 Report

In recent years, due to its exceptional oxidizing ability, ozone has been widely used in various branches of agriculture. Ozone technologies in fish farming are a promising direction. Ozonation is used to disinfect water when growing various aquatic organisms. It can be used for the prevention and treatment of fish when they are affected by helminths and parasitic protozoa. Ozonation acquires particular relevance in conditions of intensive fish breeding in recirculating water supply installations, for neutralizing water from various harmful substances.

In this regard, the relevance of this manuscript is beyond doubt.

However, there are a number of fundamental problems. The main remark concerns the absence of any statistical processing of the obtained results. Therefore, it is not clear whether the change in the studied parameters is significant and at what ozone concentration these parameters decreased relative to the control.

It is not necessary to give calculation formulas (eg phytoplankton abundance and biomass), but reference should be made to the method used. In this case, it is not clear how the phytoplankton was concentrated, which counting chamber was used, microscope, magnification....

Continuing the theme of phytoplankton. A very poor description of the dominant/discovered species of algae. Taxonomically obsolete species names are used.

I hope that the authors will be able to correct these comments.

Author Response

The results were statistically processed by adding a significance analysis.

The methods used were added to the paper, describing how phytoplankton concentrate and other processes.

Descriptions of the algal species found were added and the phytoplankton table was updated.

Round 2

Reviewer 1 Report

The authors have answered my questions and responded to my comments, and the manuscript has been improved. I recommend the publication of the manuscript in its present form.

Author Response

Thank you for your advice.
